# Deficit Irrigation at Pre-Anthesis Can Balance Wheat Yield and Water Use Efficiency under Future Climate Change in North China Plain

**DOI:** 10.3390/biology11050692

**Published:** 2022-04-30

**Authors:** Xiaoli Niu, Puyu Feng, De-Li Liu, Bin Wang, Cathy Waters, Na Zhao, Tiancheng Ma

**Affiliations:** 1College of Agricultural Equipment Engineering, Henan University of Science and Technology, Luoyang 471000, China; niuxiaoli88@126.com (X.N.); 15713993302@163.com (N.Z.); matiancheng1@gmail.com (T.M.); 2NSW Department of Primary Industries, Wagga Wagga Agricultural Institute, Pine Gully Road, Wagga Wagga, NSW 2650, Australia; 3College of Land Science and Technology, China Agricultural University, Beijing 100193, China; fengpuyu@cau.edu.cn; 4Climate Change Research Centre, University of New South Wales, High Street, Sydney, NSW 2052, Australia; 5NSW Department of Primary Industries, 34 Hampden Street, Dubbo, NSW 2830, Australia; cathy.waters@dpi.nsw.gov.au

**Keywords:** winter wheat yield, WUE, APSIM, climate change, deficit irrigation, N fertilizer rates, NCP

## Abstract

**Simple Summary:**

Adopting deficit irrigation (DI) to improve crop production and safeguard groundwater resources is of great importance in water scarce regions, e.g., the North China Plain (NCP). Under the background of global warming, it is worth investigating whether DI continues to play such a key role under future climate change scenarios. Thus, we studied the effect of DI on wheat yield and water use efficiency under future climate change scenarios. We found that moderate deficit irrigation (DI3, ≤0.4 PAWC at sowing to flowering stage) under the N3 (150 kg N ha^−1^) condition was identified as the optimum irrigation schedule for the study site under future climate change scenarios. However, the compensation effect of DI3 on yield and water use efficiency (WUE) became weak in the future. To conclude, in water scarce regions of NCP, DI remains an effective strategy to maintain higher yield and enhance water use under future climate scenarios.

**Abstract:**

Background. Deficit irrigation (DI) is a feasible strategy to enhance crop WUE and also has significant compensation effects on yield. Previous studies have found that DI has great potential to maintain crop production as full irrigation (FI) does. Therefore, adopting DI to improve crop production and safeguard groundwater resources is of great importance in water scarce regions, e.g., the North China Plain (NCP). Under the background of global warming, it is worth investigating whether DI continues to play such a key role under future climate scenarios. Methods. We studied the response of winter wheat yield and WUE to different DI levels at pre-anthesis under two Shared Socioeconomic Pathways (SSPs) scenarios (SSP245 and SSP585) using the Agricultural Production Systems Simulator (APSIM) model driven by 21 general circulation models (GCMs) from the Coupled Model Inter-Comparison Project phase 6 (CMIP6). Additionally, we explored the effects of different nitrogen (N) fertilizer application rates on DI. Results. We found that simulated wheat yield would increase by 3.5–45.0%, with WUE increasing by 8.8–46.4% across all treatments under future climate change. Moderate deficit irrigation (DI3, ≤0.4 PAWC at the sowing to flowering stage) under the N3 (150 kg N ha^−1^) condition was identified as the optimum irrigation schedule for the study site under future climate change. However, compensation effects of DI3 on yield and WUE became weak in the future, which was mainly due to increased growing season rainfall projected by GCMs. In addition, we found that N fertilizer application could mitigate the effect of DI3. Conclusions. We highlight that in water scarce regions of NCP, DI remains an effective strategy to maintain higher yield and enhance water use under future climate scenarios. Results strongly suggest that moderate deficit irrigation under a 150 kg N ha^−1^ condition could mitigate the contradiction between production and water consumption and ensure food safety in the NCP.

## 1. Introduction

Cereal crops are key stable food sources worldwide, which provide over 50% of human calorie intake directly [1]. Among them, wheat occupies a prominent position in maintaining food security [2], requiring a 60% increase by 2050 with global population growth [3]. However, wheat productivity is facing challenges from cultivated land reduction and water shortage [4,5]. To meet these challenges and ensure global food security, an integrated, multifaceted, and sustainable approach is needed to increase production per unit area and optimize resource use efficiency [6]. Furthermore, global wheat yield will be largely affected by climate change [7,8]. For example, several studies documented that each degree-Celsius increase in global mean temperature would lead to a 6–7% yield decline in many wheat growing regions of the world [9,10,11]. Therefore, it is urgent to explore the capability of different adaptation strategies to increase crop production to cope with the adverse effects of climate change, land degradation, and population explosion [12,13,14,15].

The North China Plain (NCP) is one of the main wheat growing regions in China, providing more than half of the wheat production in China. However, in this region, long-term overexploitation of groundwater has led to a decline in groundwater levels and a shortage of water resources in recent years, which has had negative impacts on sustainable development of regional agriculture [16,17,18]. Moreover, due to loss of around one-third of the water by soil evaporation, water use efficiency (WUE) in winter wheat in the region is very low. Thus, there is an urgent need to promote feasible water management strategies to maintain a high wheat productivity level in this region.

Deficit irrigation (DI) scheduling is an effective irrigation water management scheme and has been extensively applied in winter wheat in the NCP [19,20]. This is because the full irrigation (FI) approach is considered as a water luxury and is not sustainable in water scarce environments [21]. However, there are large variations in the magnitude of water use among different DI strategies. These mainly are due to the impact of the amount, duration, and timing of the imposed DI on soil water [22,23,24,25]. For example, Meena et al. [22] evaluated the water use when DI was imposed at critical wheat growth stages in 2015–2018. They reported that 75% irrigation application (45 mm irrigation amount) at all critical growth stages can save 25% of water use compared to control treatment. Xu et al. [23] reported that the highest WUE (22.7 kg ha^−1^ mm^−1^) was achieved when irrigation (60 mm) was applied only at the elongation stage in the Wuqiao experiment station of NCP in 2013–2015. Yu et al. [25] used the meta-analysis of 41 peer-reviewed publications all over the world containing over 381 observations to quantify effects of DI on wheat water use and productivity. They found that DI can improve wheat WUE by 6.6% but decreases yield by 16.2%. In addition, nitrogen (N) fertilizer application has considerable impacts on the contribution of DI to water use. For example, under DI (60% of crop evapotranspiration), N fertilizer application rates with 0.3 and 0.6 g N kg^−1^ soil greatly increased WUE by 45% and 59% compared to no-N application, respectively [2]. Thus, more comprehensive experimental designs are needed to thoroughly investigate the combined effects of DI and N application on water use and yield.

Process-based biophysical models are useful tools to simulate the interactive effects of management practices (e.g., irrigation scheduling and fertilizer application) on crop yield and water use. There are several studies using different crop models to simulate the impacts of DI on water use and yield at different sites. For example, Zhou et al. [26] applied the CERES-Wheat model to predict the winter wheat grain yield and WUE under different DI strategies using historical weather data from 33 years (1981–2014) in China. Balwinder et al. [27] used the agricultural Production Systems Simulator (APSIM) model and 40 years (1970–2010) of weather data to evaluate the interaction effects of DI, sowing data, and mulch on wheat grain yield in India. In addition, previous studies explored the potential of different DI management options under future climate change. Rashid et al. [13] used 10 general circulation models (GCM) for four representative concentration pathway (RCP) scenarios (RCP2.6, RCP4.5, RCP6.0, and RCP8.5) to assess DI impacts on wheat yield and water use with the AquaCrop model under future climate change in NCP. Ma et al. [28] simulated yield and biomass response of maize to climate change and DI with the DSSAT model under RCP8.5 with 37 GCMs in the USA. In Hale County, Texas, Kothari et al. [29] projected climate change impacts on grain sorghum yield and irrigation water use under FI and DI for RCP4.5 and RCP8.5 with nine GCMs. 

However, the majority of previous studies used future climate projections from a limited number of GCMs based on the World Climate Research Program (WCRP) of the Coupled Model Inter-Comparison Project phase 5 (CMIP5). Conversely, the latest CMIP6 provides multi-model climate projections based on alternative scenarios that are directly relevant to societal concerns regarding climate change mitigation, adaptation, or impacts [30]. CMIP6 provides less biased simulations for use in regional dynamical and statistical downscaling efforts compared to CMIP5 [31]. Notably, in CMIP6, the projected duration and intensity of drought are more robust [32]. However, no study has used the latest CMIP6 data to assess the coupled impact of climate change, different irrigation treatments, and N fertilization application on wheat yield in China. 

The major objectives of this study were to (1) calibrate and validate the APSIM model to simulate wheat yield and WUE under different management options in the NCP; (2) simulate the impacts of future climate change on wheat yield and WUE under the different combinations of representative DI and nitrogen application rates based on 21 GCMs from CMIP6 under two Shared Socioeconomic Pathways (SSPs) (SSP245 and SSP585); (3) quantify the compensation effect of DI on wheat yield and WUE under future climate change; and (4) identify optimum DI treatments under different N fertilizer rates to cope with future climate change in NCP. This study will provide insights on using promising irrigation strategies in the future to balance water use and wheat yield in the NCP.

## 2. Materials and Methods

### 2.1. Study Site, Soil Data, and Historical Climate Data

All experiments were conducted at the Qiliying Experimental Station (35°08′ N, 113°45′ E; 80.77 m), located in Xinxiang, Henan province, the center of NCP. The soil texture of the experimental station is a sandy loam, and the soil type is Haplustalf according to USDA (United States Department of Agriculture) classification, which is also the dominant soil type in the NCP. Hydrological and fertility characteristics of the soil are shown in Table 1. The study site has a continental monsoon climate with the warm season from April to September and the cool season from October to March. The winter wheat is usually sown in the middle of October and harvested in early June.

Daily values of maximum and minimum temperature (°C), sunshine duration (h), and precipitation (mm) in the study site from 1961 to 2018 were collected from a local experimental station. Daily solar radiation (MJ m^−2^) was calculated from sunshine duration according to the Angstrom formula [33]. The long-term (1961–2018) seasonal (October–June) mean temperature (Tmean), the average total rainfall, and the average total solar radiation for winter wheat were found to be 11 °C, 227 mm, and 3499 MJ m^−2^, respectively (Table 2).

### 2.2. Field Experiment Data

#### 2.2.1. Auto-Rain-Shelter Experiment in 2017–2018

In the auto-rain-shelter (three-sided rain-shelter, height of 3 m, a width of 7 m, and a length of 10 m) experiment, rainfall was blocked to remove the impact of rainfall on irrigation treatment. The experiment was conducted using sheet iron lysimeters (depth 70 cm, outer diameter 100 cm, thick plexiglass walls 1 cm) for monitoring total lysimeter weight from 2nd November 2017 to 8th June 2018. Lysimeters were held in place by stainless-steel supports. Each lysimeter bottom contained uniformly distributed holes (5 mm diameter) and was lined with filter paper. In order to keep the same surroundings as the field for the wheat, the surfaces of the lysimeters were placed flush with the ground. The soil was filled up to a height of 50 cm with a blank of 20 cm depth on the top for keeping standing water without surface runoff occurring. The local winter wheat (*Triticum aestivum* L.) cultivar “Aikang58” in the lysimeter was sown at 18 g with a row-to-row dimension of 20 cm and plant-to-plant of 6 cm. The experiment consisted of nine combinations of irrigation and N fertilizer treatments (A1–A9), and one no-N fertilizer treatment (A0) to study phenology, biomass, yield, and WUE. WUE was calculated as follows:(1)$$\mathrm{WUE}\;\left({\mathrm{kg}\;\mathrm{ha}}^{-1}{\;\mathrm{mm}}^{-1}\right)=\frac{\mathrm{yield}\;\left({\mathrm{kg}\;\mathrm{ha}}^{-1}\right)}{\;\mathrm{ET}\;\left(\mathrm{mm}\right)}$$
where seasonal evapotranspiration (ET) was calculated with the water balance equation described by Yan et al. [18]. 

The detailed irrigation treatment is shown in Table 3. Additionally, before sowing, basal fertilizers including N (50% of total N), P (90 kg ha^−1^), and K (90 kg ha^−1^) were applied in the top soil (0–40 cm). The remaining 50% of the total N was applied by surface irrigation in the returning green stage. 

#### 2.2.2. Field Experiment in 2014–2018

Another independent field experiment was conducted from 17 October 2016 to 10 June 2017 (Table 4). We used a split-plot design based on a randomized complete block designed with three replicates. The main plots consisted of two irrigation amounts (0 mm and 90 mm in the whole experiment period). The sub-plots had four N fertilizer rates (0 kg N ha^−1^, 90 kg N ha^−1^, 180 kg N ha^−1^, and 240 kg N ha^−1^). Irrigation was applied on 18th March 2017 (45 mm) and on 29th April 2017 (45 mm). N fertilizer was applied before sowing (50%) and on 15 April 2017 (50%) without irrigation, because we had some rainfall on 16th April 2017. Basal phosphorus (P_2_O_5_, 120 kg ha^−1^) and potassium (K_2_O, 105 kg ha^−1^) fertilizers were added in all treatments. The sowing density of winter wheat “Aikang58” with 20 cm row spacing was 195 kg ha^−1^. 

Additional field experimental datasets (2014–2018) were collected from independent studies conducted in the same site under different irrigation and N fertilizer treatments [34,35]. Wheat biomass, yield, and WUE were collected based on their work for different irrigation scheduling and N fertilizer rates (Table 4)

### 2.3. APSIM-Wheat Model

Here, we used APSIM version 7.7 to simulate wheat yield and water use. A detailed description of the APSIM model can be seen from http://www.apsim.info (25 January 2020). We used winter wheat “Aikang58” to calibrate the APSIM-wheat model with observed phenology, biomass, yield, and WUE data obtained from 2017–2018 (Table 4) using the trial-and-error method. The performance of the calibrated APSIM model was validated using the independent field experimental data from 2014–2018 at the same site (Table 4). The coefficient of determination (R^2^), root mean square error (RMSE), normalized root mean square error (nRMSE), and model efficiency (E) were used to assess the model performance [36].
(2)$$R^{2}=\frac{-{\left[\sum_{i=1}^{n}\left(O_{i}-\;\bar{O}\right)\left(S_{i}-\;\bar{S}\right)\right]}^{2}}{\sum_{i=1}^{n}{\left(O_{i}-\;\bar{O}\right)}^{2}\sum_{i=1}^{n}{\left(S_{i}-\;\bar{S}\right)}^{2}}$$
(3)$$\mathrm{RMSE}=\sqrt{\sum_{i=1}^{n}{\left(S_{i}-O_{i}\right)}^{2}/n}$$
(4)$$\mathrm{nRMSE}\left(\%\right)=100\times \mathrm{RMSE}/\bar{O}$$
(5)$$E=1-\frac{\sum_{i=1}^{n}{\left(O_{i}-S_{i}\right)}^{2}}{\sum_{i=1}^{n}{\left(O_{i}-\;\bar{O}\right)}^{2}}$$
where O_i_ and S_i_ are observed and simulated values, respectively, $\bar{O}$ and $\bar{S}$ are the mean of observed and simulated values, respectively, and n is the number of observed and simulated values. Generally, the smaller the RMSE, the smaller the deviation between simulated and observed values. If E is closer to 1, the model is more accurate.

### 2.4. Model Simulations

#### 2.4.1. Future Climate Data

Future climate data based on 21 GCMs (Table 5) from the CMIP6 were downloaded from https://esgf-node.llnl.gov/search/cmip6/ (20 April 2020). Monthly gridded climate data from 21 GCMs were downscaled to daily and site scale using a statistical downscaling model (NWAI-WG) [37], which has been widely used in numerous climate change impact assessment studies [12,15]. In the present study, we considered two SSPs (SSP245 and SSP585). SSP245 and SSP585 represent an intermediate “middle of the road” scenario and a high emissions “fossil-fueled development” scenario, respectively [30]. SSP585 combines the fossil-fueled development socioeconomic pathway and 8.5 Wm^−2^ forcing pathway, while SSP245 combines the moderate development socioeconomic pathway and 4.5 Wm^−2^ forcing pathway. Previous studies have already shown that atmospheric CO_2_ concentration in the APSIM-wheat model affects wheat growth by influencing RUE, TE, and leaf nitrogen concentration [12,38]. Therefore, atmospheric CO_2_ concentration is also an important variable in the wheat growth model. We used the following empirical equations to calculate the yearly atmospheric CO_2_ concentration for SSP245 and SSP585:(6)$${\;\mathrm{CO}}_{2,\mathrm{SSP}245}=62.044+\frac{34.002-3.8702Y}{0.24423-1.1542Y^{2.4901}}+0.028057{\left(Y-1900\right)}^{2}\;+0.00026827{\left(Y-1960\right)}^{3}-9.2751\times 10^{-7}{\left(Y-1910\right)}^{4}-2.2448\left(Y-2030\right)$$
(7)$${\mathrm{CO}}_{2,\mathrm{SSP}585}=757.44+\frac{84.938-1.537Y}{2.2011-3.8289Y^{-0.45242}}+2.4712\times 10^{-4}{\left(Y+15\right)}^{2}\;+1.9299\times 10^{-5}{\left(Y-1937\right)}^{3}+5.1137\times 10^{-7}{\left(Y-1910\right)}^{4}\;$$
where Y is the calendar year from 2030 to 2100 (Y = 2030, 2031, …, 2100). A constant CO_2_ concentration with 350 ppm was used for the baseline (1961–2000) simulation. We used two future time periods, namely 2030–2059 (2040s) and 2070–2099 (2080s).

#### 2.4.2. Settings for Different DI and N Fertilizer Rates

Wheat was sown at a depth of 50 mm every year once the rainfall on the 277th–293th day of the year exceeded 20 mm and harvested at physiological maturity for long-term simulations. The sowing density was set with 395 plants m^−2^. To eliminate the effect from previous seasons, simulations were reset on the sowing date of every year, including soil water, soil nitrogen content, and soil surface matter. 

Nine irrigation treatments (including seven DI treatments, FI, and rain-fed) associated with seven N fertilizer rates, a total of 63 management options, were selected to simulate the response of wheat yield and WUE to future climate change. Detailed management options can be found in Appendix A. In total, 7938 (9 irrigation treatments × 7 N fertilizer rates × 21 GCMs × 2 SSPs × 3 time periods) simulations were run based on multiple high-performance workstation computers.

### 2.5. Calculation of Future Changes in Yield, WUE Changes, and DI Compensation Effect

The equation for calculating yield and WUE changes is
(8)$$\Delta Y/\Delta \mathrm{WUE}\left(\%\right)=\frac{Y_{\mathrm{GCM}\_F}/{\mathrm{WUE}}_{\mathrm{GCM}\_F}-Y_{\mathrm{GCM}\_\mathrm{BL}}/{\mathrm{WUE}}_{\mathrm{GCM}\_\mathrm{BL}}}{Y_{\mathrm{BL}}/{\mathrm{WUE}}_{\mathrm{BL}}}\times 100$$
where ΔY/ΔWUE(%) is defined as the difference for simulated yield/WUE between the future (2040s and 2080s) and the baseline (1961–2000) periods according to GCM climate data. $Y_{\mathrm{GCM}\_F}/{\mathrm{WUE}}_{\mathrm{GCM}\_F}$ represents simulated yield/WUE for the future periods according to GCM climate data. $Y_{\mathrm{GCM}\_\mathrm{BL}}/{\mathrm{WUE}}_{\mathrm{GCM}\_\mathrm{BL}}$ indicates simulated yield/WUE for the baseline period according to GCM climate data. $Y_{\mathrm{BL}}/{\mathrm{WUE}}_{\mathrm{BL}}$ represents the simulated yield/WUE according to observed historical climate data.

We also calculated for the compensation for both yield and WUE: (9)$$Y_{\mathrm{CE}\_\mathrm{DI}}\left(\%\right)=\frac{Y_{\mathrm{DI}}-Y_{\mathrm{RN}}}{Y_{\mathrm{FI}}}\times 100$$
(10)$$WUE_{\mathrm{CE}\_\mathrm{DI}}\left(\%\right)={\frac{{\mathrm{WUE}}_{\mathrm{DI}}-{\mathrm{WUE}}_{\mathrm{RN}}}{{\mathrm{WUE}}_{\mathrm{FI}}}}_{}$$
(11)$$\Delta Y_{\mathrm{CE}\_\mathrm{DI}}/\Delta {\mathrm{WUE}}_{\mathrm{CE}\_\mathrm{DI}}=Y_{\mathrm{CE}\_\mathrm{DI}\_F}/{\mathrm{WUE}}_{\mathrm{CE}\_\mathrm{DI}\_F}-Y_{\mathrm{CE}\_\mathrm{DI}\_\mathrm{BL}}/{\mathrm{WUE}}_{\mathrm{CE}\_\mathrm{DI}\_\mathrm{BL}}\;$$
where $Y_{\mathrm{CE}\_\mathrm{DI}}$ and ${\mathrm{WUE}}_{\mathrm{CE}\_\mathrm{DI}}$ are the compensation effect for yield and WUE under DI, respectively. $Y_{\mathrm{DI}}$ is the simulated yield, and ${\mathrm{WUE}}_{\mathrm{DI}}$ is the simulated WUE for DI. $Y_{\mathrm{RN}}$ and ${\mathrm{WUE}}_{\mathrm{RN}}$ represent simulated yield and WUE under RN. $Y_{\mathrm{FI}}$ and ${\mathrm{WUE}}_{\mathrm{FI}}$ are simulated yield and WUE under FI. $\Delta Y_{\mathrm{CE}\_\mathrm{DI}}/\Delta {\mathrm{WUE}}_{\mathrm{CE}\_\mathrm{DI}}$ is the change of compensation effect of yield and WUE under DI between the future (2040s and 2080s) and the baseline (1961–2000) periods. $Y_{\mathrm{CE}\_\mathrm{DI}\_F}/{\mathrm{WUE}}_{\mathrm{CE}\_\mathrm{DI}\_F}$ is the future compensation effect for yield and WUE under DI for the future, and $Y_{\mathrm{CE}\_\mathrm{DI}\_\mathrm{BL}}/{\mathrm{WUE}}_{\mathrm{CE}\_\mathrm{DI}\_\mathrm{BL}}$ is the compensation effect for the baseline.

## 3. Results

### 3.1. Performance of the APSIM-Wheat Model and Its Parameterization

In general, the APSIM model was able to adequately simulate crop growth status of winter wheat for all ten treatments (A0–A9) (Figure 1). During the calibration phase, the model could well simulate the dynamics of phenology with an R^2^ of 0.98, an RMSE of 3.7 days, an nRMSE of 3.1%, and an E of 0.98. Moreover, the correlation coefficients (R^2^) between simulated and observed biomass, yield, and WUE were all above 0.9. The calibrated cultivar parameters for “Aikang58” are shown in Table 6.

Additionally, during the validation phase, the ASPIM model was able to explain more than 90% of the variations in phenology, biomass, and yield as well as WUE. The E for each treatment was close to 1.0 (0.91–0.96) and the simulation errors (RMSE and nRMSE) were also at a low level. All these results indicated a high degree of proficiency of the APSIM model in accurately simulating the impacts of different irrigation regimes on crop growth status of winter wheat in the study site.

### 3.2. Future Climate Projections

Figure 2 shows the projected changes in growing season Tmean, rainfall, and solar radiation in the 2040s and 2080s relative to the baseline (1961–2000) under SSP245 and SSP585 based on 21 GCMs. All GCMs indicated a warming trend for both scenarios. Specifically, the increase in Tmean under SSP585 was much greater than that under SSP245 in the 2040s and 2080s, with 1.1 °C and 2.3 °C under SSP585 and 0.9 °C and 1.4 °C under SSP245, respectively. For the growing season rainfall, SSP585 also projected a greater increase than SSP245. The ensemble-mean rainfall under SSP245 increased by 14.2% and 26.7% in the 2040s and 2080s, respectively, while the respective values for SSP585 were 17.0% and 31.9%. In addition, growing season radiation had a decrease (−1.9% and −1.2%) in the 2040s but an increase by 2.5% and 0.6% in the subsequent period under SSP245 and SSP585, respectively.

### 3.3. Projected Winter Wheat Phenology Change

Our simulation results showed that in the historical period, the days to flowering (DTF), days to maturity (DTM), and reproduction growth period (RGP) were 197.4 days, 237.8 days, and 76.4 days, respectively (Appendix A). Due to temperature increase (Figure 2), DTF and DTM were both shortened in future periods and both were shortened more under SSP585 (Figure 3). For example, DTF was reduced by 26.4 days in 2080s under SSP585 compared to 15.0 days under SSP245. On the other hand, RGP was expected to extend in the future and also was likely to extend more under SSP585 1.9 days and 8.5 days (2040s and 2080s, respectively) in comparison with 1.2 days and 3.2 days, respectively, under SSP245.

### 3.4. Projected Changes in Yield and WUE

Yield changes in 2040s and 2080s under SSP245 and SSP585 based on 21 GCMs are shown in Figure 4. Overall, for all treatments, the ensemble mean yield was projected to increase by 3.5–16.7% in the 2040s and 6.4–27.3% in the 2080s under SSP245 and 4.4–22.5% in the 2040s and 8.4–45.0% in the 2080s under SSP585. SSP585 had more promotion on yield compared to SSP245. The largest increase in yield was found in the 2080s under the SSP585 scenario.

From Figure 4, we found that there was large variation of the amplitude of the projected increase in yield among different treatments. It is noteworthy that under SSP585, RN had the largest increase in yield in the 2080s compared to other irrigation treatments, especially when the N fertilizer rates were N0 and N1. We suspect that this might be due to the greater increases in rainfall under SSP585 in the 2080s (Figure 2b). In addition, we found that regardless of N application levels, there was slight difference in projected yield among DI3–DI7 and FI (Figure 4). However, in the historical period, the variation of yield was dependent on N application levels. For example, under N0–N2, DI3–DI6 had lower yield than DI7 and FI. By contrast, under N3–N6, no significant differences were found among DI3–DI7 and FI, and we suggest that DI3–DI7 can maintain a similar yield to FI with high N application in the future.

Generally, the projected WUE change was identical to yield (Figure 5). Overall, WUE was projected to increase by 8.8–15.4% in the 2040s and 12.8–32.4% in the 2080s under SSP245 and 11.5–20% in the 2040s and 25.9–46.4% in the 2080s under SSP585. The WUE increased more for SSP585 than SSP245, especially in the 2080s. Additionally, for all irrigation treatments, the increases in WUE under N3–N6 were much greater than under N0–N2, and we suggest that an appropriate N-supply could improve WUE. When the N fertilizer rate was N3–N6, the magnitude of WUE increase under DI3–DI7 was nearly the same as FI. Based on the highest WUE value in the historical period (Appendix A) and similar increase to FI in the future period (Figure 5), DI3 would have much greater WUE values than FI. Thus, it can be identified that DI3 (≤0.4 PAWC at the sowing to flowering stage) under the N3 (150 kg N ha^−1^) condition was the optimum irrigation and N fertilizer scheduling under future climate change.

### 3.5. Wheat Yield and WUE Relationships with Future Climate

Additionally, we conducted a multiple linear regression analysis to study the impact of various climate factors on wheat yield and WUE under RN and irrigation (Table 7). Results show that under RN, Tmean and radiation have significant negative effects on yield and WUE, while rainfall, CO_2_, and soil N content have positive effects (*p* < 0.01). The R^2^ values of regression analysis were 0.81 and 0.82, respectively; therefore we think that the four climate factors and soil N content could adequately explain the wheat yield and WUE changes under RN in the future. Under irrigation, the effects of Tmean, radiation, CO_2_, and soil N content on yield and WUE were consistent with that under RN. Moreover, yield and WUE were significantly positively correlated with irrigation amount (*p* < 0.001). The R^2^ values of regression analysis were 0.77 and 0.80, respectively.

### 3.6. Projected Compensation Change of DI3 on Yield and WUE

Overall, the compensation change of DI3 on yield was similar to WUE under future climate change (Figure 6). Moreover, SSP585 had more reduction in compensation effects compared with SSP245. The yield compensation effect was decreased by 4.4% in the 2040s and 5.8% in the 2080s under SSP245 and 6.7% in the 2040s and 25.0% in the 2080s under SSP585. The WUE compensation effect was decreased by 4.3% in the 2040s and 6.1% in the 2080s under SSP245 and 7.0% in the 2040s and 27.3% in the 2080s under SSP585. In addition, the reduction of the yield compensation effect for DI3 under N3–N6 was less than under N0–N2 (except under N0 under SSP585 in the 2080s, Appendix A). Moreover, compared to N1 and N2, N3–N6 also had less reduction of WUE compensation effect for DI3 (except under N1 under SSP585 in the 2080s, Appendix A), and we feel that the mitigation effect of N fertilizer application on water deficit will still exist in the future.

## 4. Discussion

Overall, APSIM performed well to simulate wheat phenology, biomass, yield, and WUE, with the R^2^ ranging from 0.94 to 0.98, nRMSE from 2.8 to 8.7%, and E from 0.91 to 0.98 for both calibration and validation under different irrigation and N treatments (Figure 1). Previously, Sun et al. (2019) applied APSIM in NCP, and their results indicated that the model could well simulate the effects of different cropping systems under four different irrigation schedules on crop WUE and yield [39]. Our results here have confirmed the findings of Sun et al. (2019) [39] and further found that APSIM could well simulate the combined effect of control DI and N fertilizer rate on crop yield and WUE. 

Based on the simulation results, we suggest that projected climate change has positive impacts on wheat yield and WUE under rain-fed conditions (Figure 4). This can be explained as follows: GCMs project that the future temperature will increase by 0.1–0.9 °C for SSP245 and 1.4–2.3 °C for SSP585 (Figure 2), which tends to accelerate wheat growth and development leading to a shorter growth period [9]. It is interesting to note that although climate warming greatly shortens DTF and DTM (Figure 3), there is a longer RGP (0.3–12.4 day) under future climate scenarios. The prolonged grain filling period may be responsible for the yield increase. This is consistent with Xiao et al. [40] and Yan et al. [18], who reported a similar longer reproductive growth period for wheat in NCP. In addition, the increased growing season rainfall increase can partly explain the projected increase in winter wheat yield, because the total rainfall during the wheat growth period has positive impacts on yield (Table 7). Lastly, elevated CO_2_ concentration can improve wheat yield through positive impacts on wheat growth, development, and net assimilation rate [41]. This has been verified in our multiple linear regression analysis with positive correlation coefficients from CO_2_ (Table 7). Overall, increased growing season rainfall, prolonged reproductive growth period, and elevated atmospheric CO_2_ concentration counteract the negative effects of climate warming on wheat yield in our study. 

DI is a promising irrigation strategy to enhance WUE for many crops without causing a great loss of grain yield. Moreover, under drought conditions, an appropriate N supply will help crops to tolerate water deficit [2]. Therefore, it is important to explore the effects of future climate change in combination with irrigation treatments and N fertilizer rates on yield and water use. Our simulation results showed that there are interacting effects of temperature, rainfall, radiation, CO_2_, irrigation amount, and soil N content on winter wheat yield and WUE (Table 7). We identified that DI3 (≤0.4 PAWC at sowing to flowering stage) under the N3 (150 kg N ha^−1^) condition is the optimum irrigation schedule under future climate change because it obtained similar yield with high WUE as FI did (Figure 4 and Figure 5, Appendix A). Crops subjected to water deficit can compensate for some reduction in shoot growth and yield during subsequent rewatering and even maintain a similar yield under FI [42,43,44]. The physiological mechanism of the compensation effect for water deficit has already been proven by our previous study [43]. However, winter wheat has a different response to DI imposed at different growing stages [45]. Tari [46] found that the stem elongation and booting stage is more sensitive to water deficit, followed by anthesis. We found that DI3 imposed at pre-anthesis was more sensitive to increased yield and WUE under future climate change. This is possibly due to interactive effects of soil moisture, elevated CO_2_, and soil N content under future climate change (Table 7). 

Particularly, we found that the magnitude of increase in yield and WUE for DI3 under N3–N6 was greater than that under N0–N2 under future climate scenarios (Figure 4 and Figure 5). This result is consistent with the previous research that the nitrogen supplement compensates the yield loss of wheat and the deteriorative effect of water deficit conditions [47]. Physiologically, N nutrition has the potential to alleviate the drought damages by maintaining the metabolic activities even at low tissue water potential [2]. For example, under N3, the increases of yield and WUE in DI1 and DI2 were greater than RN for the majority of future climate scenarios (Figure 4 and Figure 5). Conversely, under N0, RN had slightly higher yield and WUE (Figure 5). Therefore, we propose that an appropriate N supply could stimulate crop growth, improve WUE, and alleviate the effects of drought stress [2,48], and even lead to higher increases in yield and WUE. 

However, under all N fertilizer rates, the compensation effect of DI3 on yield and WUE was largely reduced under future climate change compared to the baseline period (Appendix A). The simulated compensation effects in yield and WUE were decreased by 21.5–29.5% and 9.7–31.2% under SSP585 in the 2080s. Furthermore, we suspect that the compensation effect of DI3 on wheat will be weakening under future climate scenarios. The reason is likely due to higher temperature and more rainfall under climate change [7,18]. Climate warming will shorten the growth period, which might lead to a reduction of accumulated ET in the growth period [12]. However, due to higher seasonal rainfall in the future (Figure 2b), a substantial increase in yield under rain-fed conditions will increase WUE. Higher yield and WUE under rain-fed conditions will lead to lower value of compensation effect of supplementary irrigation, because we calculated the compensation effect by using the benchmark of rain-fed yield and WUE. Furthermore, the compensation effect of DI3 on yield and WUE would increase with the increase of N fertilizer rates. Therefore, the mitigation effect of N fertilizer application on DI would still exist under future climate scenarios. 

Long-term climate change impacts on crop yield have been widely studied using the process-based models in China [13,49,50,51] and many other countries [7,14,52,53]. Crop models are effective tools to investigate the interactive effects of climate change and different agronomic management options. Inevitably, there are some limitations involved in these simulated result. Firstly, we only considered one site (Qiliying Experimental Station) in the NCP. The growing conditions for winter wheat may change in different parts of the NCP [54]. Thus, more representative sites should be used in future studies to cover a wide range of environmental conditions including different climate and soils across the NCP. Secondly, we only used one single crop model (APSIM). The simulated yield might be overestimated because the model does not fully take into account the yield reduction resulting from diseases, pests, weeds, and extreme climate events (e.g., drought and heat stress) [15]. Furthermore, under severe deficit irrigation, the APSIM could possibly underestimate the wheat yield due to the limitations of carbohydrate mobilization mechanisms in the model [8]. Lastly, our studies only assessed the impacts of the irrigation scheduling and N fertilizer rates on wheat yield and water use under future climate change. However, other agronomic management (e.g., cultivar shift, changing sowing date, plant density, and agricultural machinery) could also play an important role in mitigating climate change [15,55,56,57]. For example, Wang et al. [15] found that breeding new cultivars and adopting earlier sowing strategies could increase yield by 20–24% under future climate change and may be particularly beneficial under dry scenarios. Even so, this study can provide useful information for future work using multiple crop models, covering different parts of the NCP, and incorporating different cultivars with changing sowing date and plant density to assess the compensation effect of deficit irrigation at a regional scale.

## 5. Conclusions

Our simulated results showed that the wheat yield and WUE are expected to increase at the study sites in the NCP under SSP245 and SSP585. We identified that moderate DI (DI3, ≤0.4 PAWC) at pre-anthesis remains a promising strategy to enhance yield and WUE under future climate scenarios. Therefore, quantification of the impacts of different DI treatments at critical growth stages on winter wheat yield and water use will assist in the development of efficient DI strategies under projected climate scenarios in the region. Furthermore, we found that the compensation effect of DI3 on yield and WUE is largely reduced under future climate change mainly due to increased growing season rainfall projected by GCMs. Therefore, it is likely that the capability of DI to cope with the adverse effects of climate change will be decreased under future climate scenarios in the NCP. In addition, our research highlights that moderate N application (N3, 150 kg Nha^−1^) could mitigate the effect of DI3. This study will provide new insights into using promising deficit irrigation and N application strategies in the future to balance water use and wheat yield in the NCP. 

## Figures and Tables

**Figure 1 biology-11-00692-f001:**
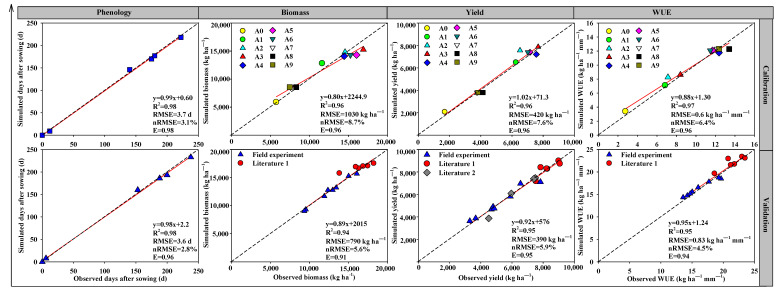
Comparison of simulated and observed values of phenology, biomass, yield, and WUE for model calibration and validation.

**Figure 2 biology-11-00692-f002:**
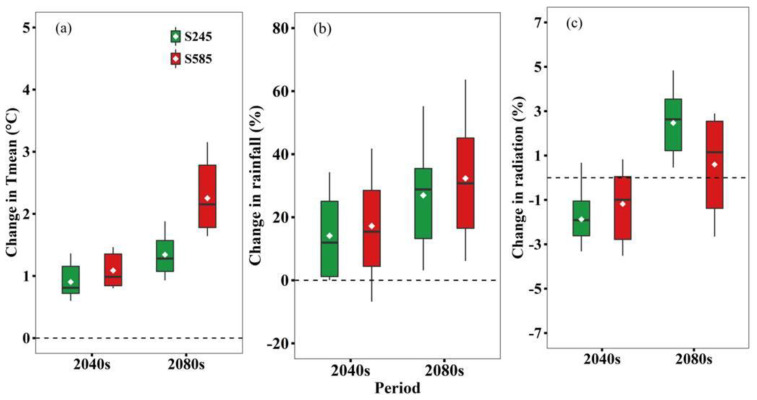
Projected changes in growing season average temperature (Tmean) (**a**), total rainfall (**b**), and total solar radiation (**c**) in 2030–2059 (2040s) and 2070–2099 (2080s) relative to the baseline (1961–2000) under SSP245 and SSP585 based on 21 GCMs. Box boundaries indicate the 25th and 75th percentiles across GCMs, and whiskers below and above the box denote the 10th and 90th percentiles. The black lines and dots inside the box indicate the multi-model median and mean, respectively.

**Figure 3 biology-11-00692-f003:**
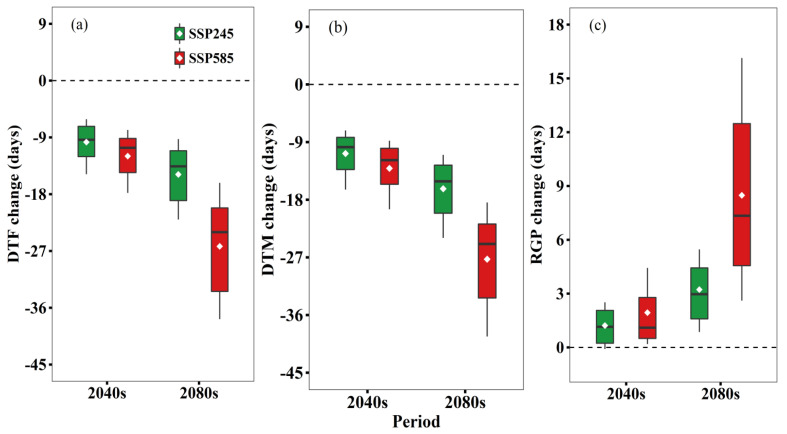
Projected changes in days to flowering (DTF) (**a**), days to maturity (DTM) (**b**), and reproduction growth period (RGP) (**c**) in 2030–2059 (2040s) and 2070–2099 (2080s) relative to the baseline (1961–2000) under SSP245 and SSP585 based on 21 GCMs. Box boundaries indicate the 25th and 75th percentiles across GCMs, and whiskers below and above the box denote the 10th and 90th percentiles. The black lines and dots inside the box indicate the multi-model median and mean, respectively.

**Figure 4 biology-11-00692-f004:**
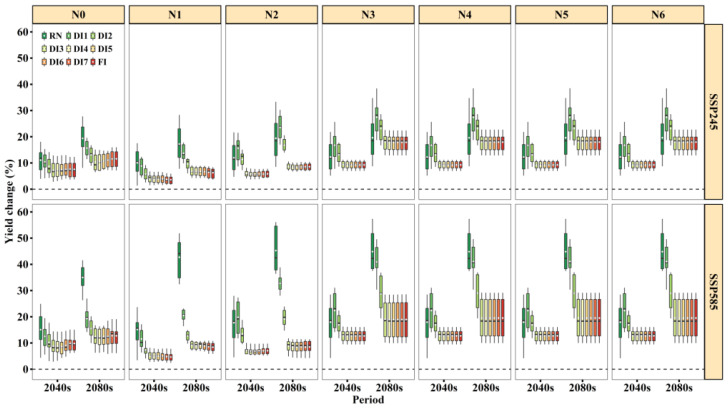
Yield change for nine irrigation treatments under N fertilizer rates in the 2040s and 2080s relative to the baseline (1961–2000) under SSP245 and SSP585 based on 21 GCMs. Box boundaries indicate the 25th and 75th percentiles across GCMs, and whiskers below and above the box denote the 10th and 90th percentiles. The black lines and white dots inside the box indicate the multi-model median and mean, respectively.

**Figure 5 biology-11-00692-f005:**
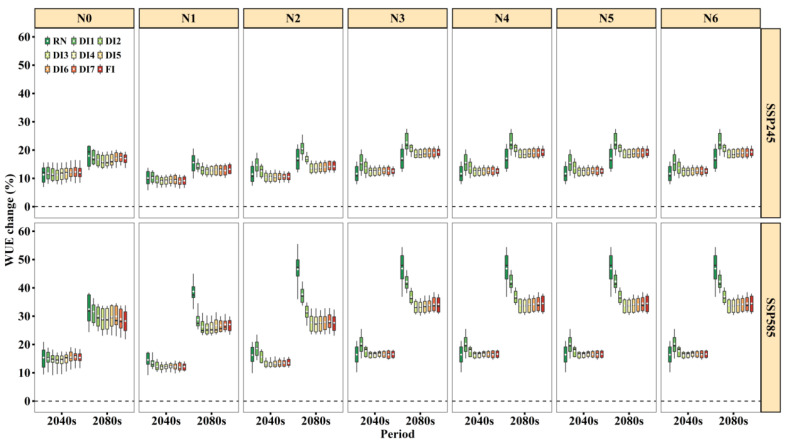
WUE change for nine irrigation treatments under N fertilizer rates in the 2040s and 2080s relative to the baseline (1961–2000) under SSP245 and SSP585 based on 21 GCMs. Box boundaries indicate the 25th and 75th percentiles across GCMs, and whiskers below and above the box denote the 10th and 90th percentiles. The black lines and white dots inside the box indicate the multi-model median and mean, respectively.

**Figure 6 biology-11-00692-f006:**
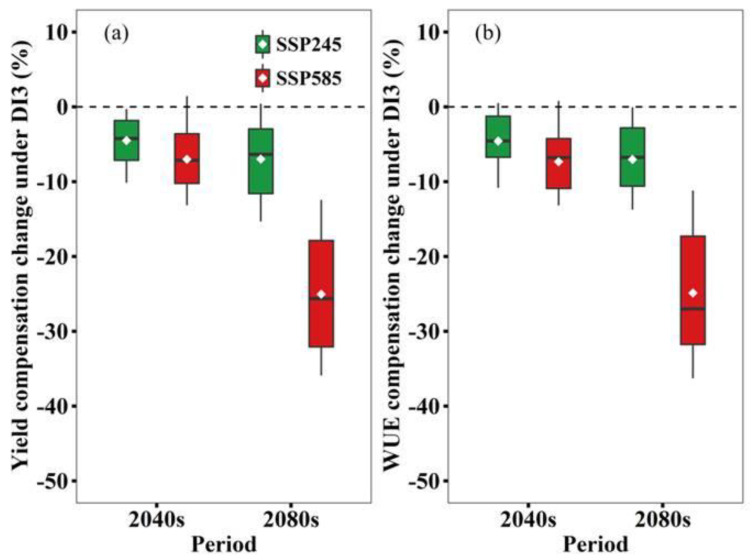
Compensation changes in yield (**a**) and WUE (**b**) under DI3 in the 2040s and 2080s relative to the baseline period under SSP245 and SSP585 based on 21 GCMs when N fertilizer rate is 150 kg ha^−1^ (N3). Box boundaries indicate the 25th and 75th percentiles across GCMs, and whiskers below and above the box denote as the 10th and 90th percentiles. The black lines and white dots inside the box indicate the multi-model median and mean, respectively.

**Table 1 biology-11-00692-t001:** The soil hydrological and fertility parameters used in the study site.

Depth (cm)	Bulk Density (g cm^−3^)	Air Dry (mm/mm)	LL15 (mm/mm)	DUL (mm/mm)	SAT (mm/mm)
0–20	1.300	0.080	0.090	0.310	0.360
20–40	1.320	0.108	0.120	0.270	0.310
40–60	1.350	0.120	0.150	0.260	0.300
60–80	1.350	0.130	0.160	0.250	0.300
80−100	1.350	0.150	0.180	0.250	0.290
100–120	1.350	0.150	0.180	0.240	0.290
120–160	1.350	0.160	0.180	0.240	0.290
160–200	1.400	0.160	0.190	0.220	0.386

Note: LL15, lower limit water content at 15 bar; DUL, drained upper limit; SAT, saturated water content.

**Table 2 biology-11-00692-t002:** Average climatic characteristics of the study area.

Month	Tmin (°C)	Tmax (°C)	Tmean (°C)	Rainfall (mm)	Radiation (MJ m^−2^)
October	10.4	21.4	15.9	31.6	349.6
November	3.4	13.9	8.7	19.8	268.0
December	−2.5	7.3	2.4	5.5	225.5
January	−4.3	5.4	0.5	4.3	238.7
February	−1.8	8.4	3.3	8.6	299.2
March	3.5	14.7	9.1	16.2	410.7
April	9.9	21.4	15.7	32.4	514.5
May	15.5	27.2	21.4	42.9	593.8
June	20.2	31.8	26.0	65.6	599.1

Note: Tmin, Tmax, and Tmean indicate mean monthly minimum, maximum, and mean temperature from 1961 to 2018, respectively. Rainfall and radiation indicate total monthly rainfall and radiation from 1961 to 2018, respectively.

**Table 3 biology-11-00692-t003:** Detailed irrigation and fertilization treatments for the auto-rain-shelter experiment.

Treatments	Irrigation Treatments under Controlled Deficit Irrigation			N Fertilizer Rates (kg N ha^−1^)
	Sowing to Flowering Stage	Flowering to Grain Filling Stage	Grain Filling Stage to Maturity	
A0	60–75%	75–100%	65–100%	0
A1	80–95%	75–100%	65–100%	240
A2	80–95%	75–100%	65–100%	300
A3	80–95%	75–100%	65–100%	360
A4	60–75%	75–100%	65–100%	240
A5	60–75%	75–100%	65–100%	300
A6	60–75%	75–100%	65–100%	360
A7	50–65%	75–100%	65–100%	240
A8	50–65%	75–100%	65–100%	300
A9	50–65%	75–100%	65–100%	360

Note: 60–75% indicates that when soil water is less than 60%, irrigation was applied until soil water content reached 75%. Other values have the same mean as 60–75% in the table.

**Table 4 biology-11-00692-t004:** Detailed experimental and literature information about wheat cultivar “Aikang58” for model calibration and validation.

Subset	Data Source	Sowing Date	Harvest Date	Treatments	Observed Data
Calibration	Auto-rain-shelter experiment (2017–2018)	2 November 2017	8 June 2018	A0–A9 (see Table 3)	Phenology, biomass, yield, WUE
Validation	Field experiment 1 (2016–2017)	17 October 2016	10 June 2017	N1: 240 kg N ha^−1^, 90 mm IA	Phenology, biomass, yield, WUE
				N2: 180 kg N ha^−1^, 90 mm IA	
				N3: 90 kg N ha^−1^, 90 mm IA	
				N4: 0 kg N ha^−1^, 90 mm IA	
				N5: 240 kg N ha^−1^, 0 mm IA	
				N6: 180 kg N ha^−1^, 0 mm IA	
				N7: 90 kg N ha^−1^, 0 mm IA	
				N8: 0 kg N ha^−1^, 0 mm IA	
	Field experiment2 (2014−2016) (Kumar Jha et al., 2019)	18 October 2014	6 June 2015	F1: FI: 50% of FC; TIA: 120 mm	Biomass, yield, WUE
				F2: FI: 60% of FC; TIA: 180 mm	
				F3: FI: 70% of FC; TIA: 240 mm	
		15 October 2015	3 June 2016	F1: FI: 50% of FC; TIA: 120 mm	
				F2: FI: 60% of FC; TIA: 180 mm	
				F3: FI: 70% of FC; TIA: 240 mm	
	Field experiment 3 (2017–2018) (Zhao et al., 2020)	15 October 2017	10 October 2018	N0: 0 kg N ha^−1^; N100: 100 kg N ha^−1^; N200: 200 kg N ha^−1^;	Yield
				N300: 300 kg N ha^−1^	

Note: IA: irrigation amounts; FI: flooding irrigation lower limit; FC: field capacity; TIA: total irrigation amount.

**Table 5 biology-11-00692-t005:** List of 21 general circulation models (GCMs) under SSP245 and SSP585 future climate scenarios used in the study for statistical downscaling outputs of the Qiliying experimental station in Xinxiang City, Henan Province, China.

Model ID	Name of GCM	Abbr. of GCM	Institute ID	Country
01	ACCESS–CM2	ACM	CSIRO–BOM	Australia
02	ACCESS–ESM1–5	AE5	CSIRO–BOM	Australia
03	BCC–CSM2–MR	BCM	BCC	China
04	CanESM5	Ca5	CCCMA	Canada
05	CanESM5–CanOE	CaC	CCCMA	Canada
06	CNRM–CM	CCM	CNRM	France
07	CNRM–ESM	CES	CNRM	France
08	EC–Earth3	EE3	EC–EARTH	Europe
09	EC–Earth3–Veg	EEV	EC–EARTH	Europe
10	FGOALS–g3	FG3	FGOALS	China
11	GFDL–ESM4	GE4	NOAA GFDL	USA
12	GISS–E2–1–G	GEG	NASA GISS	USA
13	INM–CM5–0	IC0	INM	Russia
14	INM–CM4–8	IC8	INM	Russia
15	IPSL–CM	ICM	IPSL	France
16	MIROC6	MC6	MIROC	Japan
17	MIROC–ES2L	ME2	MIROC	Japan
18	MPI–ESM1–2–HR	MEH	MPI–M	Germany
19	MPI–ESM1–2–LR	MEL	MPI–M	Germany
20	MRI–ESM	MEM	MPI–M	Germany
21	UKESM1–0–LL	U0L	NCAS	UK

**Table 6 biology-11-00692-t006:** Genetic parameters for winter wheat cultivar “Aikang58” in this study.

Name	Definition	Unit	Aikang58
photop_sens	Photoperiod sensitivity	−	3.5
vern_sens	Vernalization sensitivity	−	2
tt_end_of_juvenile	Thermal time from sowing to end of the juvenile	°C day	570
startgf_to_mat	Thermal time from beginning of grain-filling to maturity	°C day	580
tt_floral_initiation	Thermal time from floral initiation to flowering	°C day	570
tt_start_grain_fill	Thermal time from the start of grain filling to maturity	°C day	700
max_grain_size	Maximum grain size	g	0.047
potential_grain_filling rate	Potential daily grain filling rate	g grain^−1^ day^−1^	0.004
grains_per_gram_stem	Grain number per stem weight at the start of grain filling	g	25
y_frac_leaf	Fraction of remaining dry matter allocated to leaves	−	0.3
x_stem_wt	Stem weight per plant	g/plant	6
y__height	Plant canopy height	mm	1500

**Table 7 biology-11-00692-t007:** The coefficients of the regression analysis (a, b, c, d, e, f, F_0_) for assessing the impact of climate change on wheat yield and WUE under rain-fed (RN) and irrigation. ΔY/ΔWUE = aΔT + bΔP + cΔR + dΔCO_2_ + eΔI + fΔSN + F_0_.

Treatment	Output Indicator	a	b	c	d	e	f	F_0_	R^2^
RN	ΔY	−217.6 ***	18.5 ***	−62.8 ***	357.7 ***	−	0.84 ***	−5050 ***	0.81
	ΔWUE	−0.70 ***	0.01 **	−0.20 ***	1.10 ***	−	0.02 ***	−13.2 ***	0.82
Irrigation	ΔY	−53.7 **	−	−16.9 ***	177.6 ***	6.77 ***	22.7 ***	320.2 ***	0.77
	ΔWUE	−0.46 ***	−	−0.10 ***	0.58 ***	0.002 ***	0.03 ***	0.46 ***	0.80

Note: The change in simulated yield (ΔY, kg ha^−1^) and WUE (ΔWUE, kg ha^−1^ mm^−1^) as functions of the change in growth period mean temperature (ΔT, °C), rainfall (ΔP, %), solar radiation (ΔR, %), CO_2_ concentration (ΔCO_2_, 100 ppm), irrigation amount (ΔI, mm), and soil N content (ΔSN, kg ha^−1^) are shown in the multiple linear regression model. ** and *** indicate the significant at the level of *p* < 0.01 and *p* < 0.001, respectively.

## Data Availability

Not applicable.

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
