# Peer review of "Deficit Irrigation at Pre-Anthesis Can Balance Wheat Yield and Water Use Efficiency under Future Climate Change in North China Plain"

_biology, 2022, doi:10.3390/biology11050692_

Round 1

Reviewer 1 Report

the paper presents a simulation exercise on wheat yields in the future climate, with reference to a North China Plain experimental station. the paper refers to a semi-arid climate and deals with deficit emergency and its interaction with nitrogen fertilization. I don’t understand the motivation that led to make these simulations, while climate change projections foresee an increase in rainfall. in fact the result  found is that yields will increase regardless of the quantities of water given for irrigation. that said, the article is well set up and there is a good calibration / validation of the modeling. Is the tool you put in place suitable  for applications on other sites, where does talking about emergency irrigation have a reason to exist?

That said, the analysis of the limitation of your research were appreciated (lines 238-258).

In any case, the paper needs some minor revision:

Pag.4: are the barrels a kind of lysimeter?  Detail and describe better

Page 6, 7, page 9 etc: the formulas are poorly written, write them more clearly

Pag 6-7: why do you calculate so roughly the CO2 of the future when it is made available by the IPCC? Check the value obtained and correct the simulations if necessary.

Author Response

  1. I don’t understand the motivation that led to make these simulations, while climate change projections foresee an increase in rainfall.

Response: Thank you very much for your support and valuable comments. Actually,deficit irrigation (DI) has been extensively applied in winter wheat in the NCP. However, there are large variations in the magnitude of water use due to the impact of the amount, duration and timing of the imposed DI on soil water. In the auto-rain-shelter experiment, we found that DI has great potential to maintain crop production as full irrigation (FI) does. And DI degree during sowing to flowering stage has a significant influence on wheat phenology, biomass, yield and WUE. Therefore, under the background of global warming, it is worth investigating whether DI continues to play such a key role under future climate scenarios. In addition, nitrogen (N) fertilizer application has considerable impacts on the contribution of DI to water use. This study will provide new insights on using promising deficit irrigation and N application strategies in the future to balance water use and wheat yield in the NCP.

  1. Pag.4: are the barrels a kind of lysimeter? Detail and describe better

Response: Yes. The barrels are a kind of lysimeter. We have carefully described the barrels. The experiment was conducted using sheet iron lysimeters (depth 70 cm, outer diameter 100 cm, thick plexiglass walls 1 cm) for monitoring total lysimeter weight during 2nd November 2017 and 8th June 2018. Lysimeters were held in place by a stainless steel supports. Each lysimeter bottom contained uniformly distributed holes (5 mm diameter) and was lined with filter paper. In order to keep the same surroundings as the field for the wheat, the surface of the lysimeters were placed flush with the ground. The soil was filled up to a height of 50 cm with a blank of 20 cm depth on the top for keeping standing water without occurring for surface runoff. Local winter wheat (Triticum aestivum L.) cultivar “Aikang58” in the lysimeter was sown at 18 g with a row-to-row dimension of 20 cm and plant-to-plant of 6 cm.

  1. Page 6, 7, page 9 etc: the formulas are poorly written, write them more clearly

Response: Thank you very much for your suggestion. The formulas had been spaced with the text and been formatted as an equation using an equation editor.

  1. Pag 6-7: why do you calculate so roughly the CO2 of the future when it is made available by the IPCC? Check the value obtained and correct the simulations if necessary.

Response: To simulate the responses in these parameters referring to [CO2] in APSIM, we used the same approaches as adopted for CMIP3 (Yang et al., 2014) and CMIP5 (Liu et al., 2017) by fitting the IPCC reported [CO2] values to an empirical equation for calculation of the yearly [CO2] required by APSIM simulation. In our study, we fitted the [CO2] values for the ssp245 and SSP585 scenarios of CMIP6 obtained from O'Neill et al. (2016) to the following empirical equation.

Yang, Y., Liu, D.L., Anwar, M.R., et al. 2014. Impact of future climate change on wheat production in relation to plant-available water capacity in a semiarid environment. Theor. Appl. Climatol. 115, 391–410.

Liu, D.L., Zeleke, K.T., Wang, B., et al. 2017. Crop residue incorporation can mitigate negative climate change impacts on crop yield and improve water use efficiency in a semiarid environment. Eur. J. Agron. 85, 51–68.

O’Neill, B.C., Tebaldi, C., van Vuuren, D.P., et al. 2016. The scenario model intercomparison project (ScenarioMIP) for CMIP6. Geosci. Model Dev. 9, 3461–3482.

Reviewer 2 Report

First of all, my congratulations on the work.  Although it is a good paper, authors must work more on the way of transmitting it. It was a very tedious reading and it is necessary to improve the presentation and organize the article in relation to figures.

Step to detail some considerations:

  1. Simple summary

Line 19. Climate change scenarios

Line 23. It is not correct to use the acronym (WUE) if in line 20 it is not left defined first.

  1. Introduction

Line 75. “Dis” is not correct if it is not defined before. To talk about deficit irrigation strategies in a plural way, it is correct to say "DI strategies" and be consistent with the use of language.

Line 80. kg ha-1 mm-1

  1. Materials and methods

Line 131. The word “Altitude” is not necessary.

Line 132. The authors talk about a "sandy loam" soil. But under what classification? Cite the classification followed for the characterization of the type of soil.

Line 133. “The soil hydrological and fertility are shown in Table 1” This sentence is incomplete. Perhaps the authors meant: “Hydrological and fertility characteristics of the soil are shown in Table 1”.

Line 137. Table 1. Leave a title on the table and use the table footer to explain acronyms.

Line 139-141. Incongruous sentence. It must be rewritten.

Line 146. Table 2. All tables need a title and a table footer. The table footer cannot be used as a title. "Average climatic characteristics of the study area”, and in the footer explain the concepts.

Line 150. Why the rainfall was blocked? Explain it.

Line 153. The scientific name of the species must be in italics.

Line 158. This comment is general for all the equations in the article. They must be spaced with the text and must be formatted as an equation, not plain text. Every text editor program has an equation editor, use it.

Line 161. “The detailed irrigation treatment […]”

Line 165. Table 3. There is a misconception between deficit irrigation strategies and deficit irrigation treatments. The table presents deficit irrigation treatments applied in its deficit irrigation strategy. The authors do not explain what kind of strategy they carry out. Is it controlled deficit irrigation or sustained deficit irrigation? explain it, please.

Line 183. Table 4. It is necessary to look for another way of showing what Table 4 explains, since the same words are repeated a lot. Either another way of explaining this information is proposed or acronyms such as "Irrigation amounts (IA)" or "Total irrigation amount (TIA)".

Line 193. R2

Line 228. Missing space between SSP585 and future.

Line 237. The authors must be consistent with the language, if they have already defined DI and FI they must use it.

Line 240. Missing space between 7938 and parenthesis.

Table 5. If they use bold format for the heading, they must use it in all.

I don't know why but on page 9 the line numbering starts again at 1.

  1. Results

Line 25, 26, 117 and 122. R2

Line 116 and 121. the "p" must be in italics and lowercase (p<0.01).

Line 123. F0

Table 7. Table 7 has an excessively long title. The table title should not be used to explain concepts, that is why the table footer is used.

Line 136-138. The phrase "Based on our results, [...]" should go to discussion not in the results part.

  1. Discussion

I think it is not correct to present figures in the discussion section and that these figures do not have a consistent numbering with the results.

From my point of view, all the figures should go to results and in the discussion make references to the figures.

  1. Conclusions

Line 262. Use DI

In general, a good topic of research work with novel results is presented. However, the way of presenting it must be significantly improved and make the text more pleasant when reading it.

Author Response

First of all, my congratulations on the work. Although it is a good paper, authors must work more on the way of transmitting it. It was a very tedious reading and it is necessary to improve the presentation and organize the article in relation to figures.

Response: Thank you for your suggestion. We have carefully read and revised the manuscript, eg. Table S1, Figure S1, Figure S2, Figure S3, Figure S4, Figure S5, were moved into the section of supplementary materials. All changes are marked in red color in the main text, and a point-by-point response is as follows:

 Detailed comments:

  1. Line 19. Climate change scenarios

Response: Thank you very much for your suggestion. Replaced climate scenarios with Climate change scenarios.

  1. Line 23. It is not correct to use the acronym (WUE) if in line 20 it is not left defined first.

Response: WUE was defined first in line 23.

  1. Line 75. “Dis” is not correct if it is not defined before. To talk about deficit irrigation strategies in a plural way, it is correct to say "DI strategies" and be consistent with the use of language.

Response: We accepted the suggestion of Reviewer. Replaced Dis with DI strategies.

  1. Line 80. kg ha-1 mm-1 

Response: Thank you very much for your suggestion. kg ha-1 mm-1 was revised as kg ha-1 mm-1.

  1. Line 131. The word “Altitude” is not necessary.

Response: We delete it.

  1. Line 132. The authors talk about a "sandy loam" soil. But under what classification? Cite the classification followed for the characterization of the type of soil.

Response: We accepted the suggestion of Reviewer. The sentence had been revised as “The soil texture of the experimental station is a sandy loam and the soil type is Haplustalf according to USDA (United States Department of Agriculture) classification, which is also the dominant soil type in the NCP.”

  1. Line 133. “The soil hydrological and fertility are shown in Table 1” This sentence is incomplete. Perhaps the authors meant: “Hydrological and fertility characteristics of the soil are shown in Table 1”.

Response: We accepted the suggestion of Reviewer. The sentence had been revised as “Hydrological and fertility characteristics of the soil are shown in Table 1” .

  1. Line 137. Table 1. Leave a title on the table and use the table footer to explain acronyms.

 Response: We accepted the suggestion of Reviewer, and revised table 1.

  1. Line 139-141. Incongruous sentence. It must be rewritten.

Response: The sentence was rewritten as “Daily values of maximum and minimum temperature (°C), sunshine duration (h) and precipitation (mm) in the study site from 1961−2018 were collected from a local experimental station.”

  1. Line 146. Table 2. All tables need a title and a table footer. The table footer cannot be used as a title. "Average climatic characteristics of the study area”, and in the footer explain the concepts.

 Response: We accepted the suggestion of Reviewer, and revised table 2.

  1. Line 150. Why the rainfall was blocked? Explain it.

Response: In the auto-rain-shelter experiment, the objective was to evaluate the wheat water use and yield when deficit irrigation degree was imposed at critical growth stages. Therefore, the rainfall was blocked to avoid the its effect on soil water content.

  1. Line 153. The scientific name of the species must be in italics.

Reponse: Yes. The scientific name of the species had been revised.

  1. Line 158. This comment is general for all the equations in the article. They must be spaced with the text and must be formatted as an equation, not plain text. Every text editor program has an equation editor, use it.

Response: Thank you very much for your suggestion. The formulas had been spaced with the text and been formatted as an equation using an equation editor.

  1. Line 161. “The detailed irrigation treatment […]”  

Response: Replaced The detailed treatment with The detailed irrigation treatment.

  1. Line 165. Table 3. There is a misconception between deficit irrigation strategies and deficit irrigation treatments. The table presents deficit irrigation treatments applied in its deficit irrigation strategy. The authors do not explain what kind of strategy they carry out. Is it controlled deficit irrigation or sustained deficit irrigation? explain it, please.

Response: It was revised as “Irrigation treatments under controlled deficit irrigation”.

  1. Line 183. Table 4. It is necessary to look for another way of showing what Table 4 explains, since the same words are repeated a lot. Either another way of explaining this information is proposed or acronyms such as "Irrigation amounts (IA)" or "Total irrigation amount (TIA)".

Response: we agree with the suggestion of reviewer, in revised manuscript, we use acronyms to avoid the question.

  1. Line 193. R2

Response:Have revised it.

  1. Line 228. Missing space between SSP585 and future.

Response: We added the space between SSP585 and future.

  1. Line 237. The authors must be consistent with the language, if they have already defined DI and FI they must use it.

Response: Have revised it.

  1. Line 240. Missing space between 7938 and parenthesis.

Response: We added the space between 7938 and parenthesis.

  1. Table 5. If they use bold format for the heading, they must use it in all.

Response: we use bold format for the heading in table 5.

  1. I don't know why but on page 9 the line numbering starts again at 1.

Response: Have revised it.

Results

Line 25, 26, 117 and 122. R2

Response: Have revised it.

Line 116 and 121. the "p" must be in italics and lowercase (p<0.01).

Response: the "p" have been revised in italics and lowercase.

Line 123. F0

Response: Have revised it.

Table 7. Table 7 has an excessively long title. The table title should not be used to explain concepts, that is why the table footer is used.

Response: we have revised it and added the table footer.

Line 136-138. The phrase "Based on our results, [...]" should go to discussion not in the results part.

Response: The phrase "Based on our results, [...]"have been deleted.

Discussion

I think it is not correct to present figures in the discussion section and that these figures do not have a consistent numbering with the results.

Response: we agree with the suggestion of reviewer, in revised manuscript, we delete figures in the discussion section.

From my point of view, all the figures should go to results and in the discussion make references to the figures.

Response: we have carefully checked and deleted figures in the discussion section.

Conclusions

Line 262. Use DI

Response: Have revised it.

Reviewer 3 Report

Comments

SUMMARY

The paper addresses the research area related to irrigation of wheat and its biology in the MDPI Biology journal. I believe that the target journal is an appropriate forum for this article. The major objectives of this study were to (i) calibrate and validate the APSIM model to simulate wheat yield and WUE under different management options in the NCP; (ii) simulate the impacts of future climate change on wheat yield and WUE under the different combinations of representative DI and nitrogen application rates based on GCMs from CMIP6 under two Shared Socioeconomic Pathways (SSPs) (SSP245 and SSP585); (iii) quantify the compensation effect of DI on wheat yield and WUE under future climate change, and (iv) identify optimum DI strategies under different N fertilizer rates to cope with future climate change in NCP.

BROAD COMMENT

This study is of great importance for wheat production in China. The Introduction section is written with recent references. The methods were well described and in detail allowing a good understanding of the results of the study. They discussed well the results of the study. However, I have some concerns about the different parts of the manuscript. I suggest a major revision to address a few issues. If the authors address carefully the comments, I’ll recommend publication of the manuscript in the journal.

SPECIFIC COMMENTS

  • Figure S1: For DMT phenology, on top of the bar it is written 274, but in reality, considering the y-axis scale, it shows something around 240. Please, do check this and correct it.
  • Figure 1: is not clear and difficult to read what is written there. Please, do improve the quality of Figure 1.
  • Please, move Table 5 to the results section.
  • Table 5: Some parameters appear most of the time in the calibration of APSIM that seems to be missing in your study. These are x_frac_leaf (Vector with phenological stages for which y_frac leaf contains acorresponding value), y_frac_leaf (Vector with phenology dependent fractions of newly produced biomass partitioned to leaves), x_stem_wt (Vector with phenological stages for which y_stem_wt contains a corresponding value) and y_stem_height (Vector with stem weight dependent heights). Please, can you explain this?
  • Lines 22-33: The authors failed to compare the results of the performance of their calibrated APSIM wheat with previous similar studies. The discussion is missing, please do include it.
  • Lines 260-272: The authors failed to include the implications of the findings of the study for wheat production in the North China Plain. Please, do include this as well in the abstract.

Author Response

Figure S1: For DMT phenology, on top of the bar it is written 274, but in reality, considering the y-axis scale, it shows something around 240. Please, do check this and correct it.

Response: Thank you for your suggestion. We have carefully read and check figure S1. For DMT phenology, on top of the bar it is rewritten as 238.

Figure 1: is not clear and difficult to read what is written there. Please, do improve the quality of Figure 1.

Response: Thank you very much for your suggestion. We improve the quality of Figure 1.

Please, move Table 5 to the results section.

Response: Yes. We did it.

Table 5: Some parameters appear most of the time in the calibration of APSIM that seems to be missing in your study. These are x_frac_leaf (Vector with phenological stages for which y_frac leaf contains acorresponding value), y_frac_leaf (Vector with phenology dependent fractions of newly produced biomass partitioned to leaves), x_stem_wt (Vector with phenological stages for which y_stem_wt contains a corresponding value) and y_stem_height (Vector with stem weight dependent heights). Please, can you explain this?

Response: In table 5, we added some parameters eg. y_frac_leaf, x_stem_wt, y_ height.

Lines 22-33: The authors failed to compare the results of the performance of their calibrated APSIM wheat with previous similar studies. The discussion is missing, please do include it.

Response: In the discussion, we included it. “Overall, APSIM performed well to simulate wheat phenology, biomass, yield and WUE with the R2 ranging from 0.94 to 0.98, nRMSE from 2.8 to 8.7% and E from 0.91 to 0.98 for both calibration and validation under different irrigation and N treatments (Figure 1). Previously, Sun et al. (2019) applied APSIM in NCP, and their results indicated that the model could well simulated the effects of different cropping systems under four different irrigation schedules on crop WUE and yield [39]. Our results here have confirmed the findings of Sun et al. (2019) [39] and further found that APSIM could well simulate com-bined effect of control DI and N fertilizer rate on crop yield and WUE.”

Lines 260-272: The authors failed to include the implications of the findings of the study for wheat production in the North China Plain. Please, do include this as well in the abstract.

Response: In abstract, we added the sentence “Results strongly suggest that Moderate deficit irrigation under 150 kg N ha−1 condition could mitigate the contradiction between production and water consumption and ensure the food safety in the NCP.”

Round 2

Reviewer 2 Report

The work has significantly improved its presentation and content.

Congratulations

Reviewer 3 Report

I have undertaken a review of the manuscript (revised) as well as the attached author responses to the initial review. I am satisfied with the revisions made by the authors as they have addressed most, if not all, of my initial comments. Therefore, I do believe that the manuscript has been significantly improved and now warrants publication in Biology.
